# QTRAP LC/MS/MS of Garlic Nanoparticles and Improving Sunflower Oil Stabilization during Accelerated Shelf Life Storage

**DOI:** 10.3390/foods11243962

**Published:** 2022-12-07

**Authors:** Nouara Abdelli, Enas Mekawi, Mohammed Ebrahim Abdel-Alim, Nesreen Saad Salim, Mahran El-Nagar, Sati Y. Al-Dalain, Ridab Adlan Abdalla, Ganesan Nagarajan, Emad Fadhal, Rashid I. H. Ibrahim, Eman Afkar, Mohamed K. Morsy

**Affiliations:** 1Department of Basic Sciences, King Faisal University, P.O. Box 400, Al-Ahsa 31982, Saudi Arabia; 2Department of Agricultural Biochemistry, Faculty of Agriculture, Benha University, Moshtohor, Qaluobia P.O. Box 13736, Egypt; 3Department of Horticulture, Faculty of Agriculture, Benha University, Moshtohor, Qaluobia P.O. Box 13736, Egypt; 4Department of Medical Support, Al-Karak University College, Al-Balqa Applied University, Salt P.O. Box 19117, Jordan; 5Department of Mathematics & Statistics, College of Science, King Faisal University, P.O. Box 400, Al-Ahsa 31982, Saudi Arabia; 6Department of Biological Sciences, College of Science, King Faisal University, P.O. Box 400, Al-Ahsa 31982, Saudi Arabia; 7Department of Botany, Faculty of Science, Khartoum University, P.O. Box 321, Khartoum 11115, Sudan; 8Department of Botany and Microbiology, College of Science, Bani-Suef University, Bani-Suef P.O. Box 52621, Egypt; 9Department of Food Technology, Faculty of Agriculture, Benha University, Moshtohor, Qaluobia P.O. Box 13736, Egypt

**Keywords:** garlic, nanoparticles, sunflower oil, shelf life, QTRAP LC/MS/MS, accelerated storage

## Abstract

The purpose of this research was to assess and utilize the bioactive compounds of garlic nanoparticles (Ga-NPs) as a natural antioxidant in sunflower oil (SFO) stored at 65 ± 1 °C for 24 days. The garlic nanoparticles (Ga-NPs) from the Balady cultivar were prepared, characterized, and added to SFO at three concentrations: 200, 600, and 1000 ppm (*w*/*v*), and they were compared with 600 ppm garlic lyophilized powder extract (Ga-LPE), 200 ppm BHT, 200 ppm α-tocopherol, and SFO without Ga-NPs (control). The QTRAP LC/MS/MS profile of Ga-NPs revealed the presence of four organosulfur compounds. Ga-NPs exhibited the highest capacity for phenolic, flavonoid, and antioxidant compounds. In Ga-NP SFO samples, the values of peroxide, *p*-anisidine, totox, conjugated dienes, and conjugated trienes were significantly lower than the control. The antioxidant indices of SFO samples containing Ga-NPs were higher than the control. The Ga-NPs enhanced the sensory acceptability of SFO treatments up to day 24 of storage. The shelf life of SFO treated with Ga-NPs was substantially increased (presuming a Q_10_ amount). The results show that Ga-NPs are a powerful antioxidant that improves SFO stability and extends the shelf life (~384 days at 25 °C).

## 1. Introduction

Sunflower oil (SFO) is a popular vegetable oil that is commonly consumed all over the world [1]. SFO contains a high concentration of polyunsaturated fatty acids (PUFAs), particularly linoleic acid (C18:2, 68–72%) [2]. SFO is more sensitive to oxidative degradation, resulting in rancid odors, off flavors, and discoloration [3]. To improve oil stability, artificial antioxidants such as BHT (butylated hydroxytoluene) and TBHQ (tertiary butylhydroquinone) have been used as additives [4]. However, at high doses (>200 ppm), these compounds cause numerous health risks, including cancer [5]. Recently, the uses of natural antioxidants extracted from plants have received a lot of attention [6], such as essential oil [7] and sesame peel [8]. Natural antioxidants have numerous advantages, including being generally recognized as safe (GRAS), healthy, thermally stable, and readily available, as well as there being a consumer preference for ‘greener’ labels that state “no synthetic food additives”. Garlic (*Allium sativum* L.) is a large economic vegetable and medicinal plant in Egypt [9]. Garlic, from the Liliaceae family, is well-known for its high content of sulfur compounds such as γ-glutamyl peptides: γ-L-glutamyl-S-allyl-L-cysteine (GSAC), γ-L-glutamyl-S-methyl-L-cysteine (GSMC), γ-L-glutamyl-S-(trans-1-propenyl)-L-cysteine (GSPC), and γ-glutamylphenylalanine (cGPA) [10]. It is a reliable source of antioxidants, including phenolic, flavonoid, and allicin compounds [11], as well as antimicrobial [12] and anticancer properties [13]. Therefore, garlic is used as a food additive to prevent rancidity in meat [14] and for oil protection [15]. Garlic’s chemical structure is influenced by a number of factors, such as agronomic conditions, environmental conditions, and cultivars [16]. Nanotechnology is a relatively recent approach with significant potential in the food industry [17], which includes: (i) enhancing food biosafety [18], (ii) increasing shelf life [19], and (iii) active packaging [20]. One study by Morsy, Elbarbary and Saad [6] found that spirulina nanoparticles slowed lipid oxidation in olive oil during storage. Another investigation by Shehzad, et al. [21] found that nano-encapsulation-based curcumin has antioxidant activity in fish oil. Previous research attempted to improve the storage stability of sunflower oil by using natural antioxidants such as potato peels [22], grape seed [23], rosemary extract [24], and thyme extract [25]. However, no research has been conducted on the use of garlic nanoparticles (Ga-NPs) as a recent antioxidant for improving the stability of edible oil. Thus, the primary objective of the study was to assess (i) the antioxidant potential of various garlic cultivars and nanoparticles (Ga-NPs) and (ii) the effect of Ga-NPs on the storage of sunflower oil under expedited oxidation.

## 2. Materials and Methods

### 2.1. Raw Materials and Reagents 

Sunflower oil (SFO) was supplied from Arma Food Industries Company (Cairo, Egypt). A total of five fresh cultivars of garlic (*Allium sativum* L.), i.e., Egyptian garlic (Balady), Italian Red, Sids-40, Chinese, and Brazilian Hozan, that were in season (2021–2022) were obtained from the Horticulture Research Institute (Cairo, Egypt). Folin–Ciocalteu reagent, 2,2-diphenyl-1-picrylhydrazyl (DPPH), 2,2′-azinobis (3-ethylbenzothiazoline)-6-sulfonic acid (ABTS), butylated hydroxytoluene (BHT), α-tocopherol, gallic acid, and rutin were procured from Sigma-Aldrich Co. (Germany). All chemicals were of HPLC-grade and obtained by Merck (Germany). 

### 2.2. Preparation of Garlic Nanoparticles (Ga-NPs) and Garlic Extracts

Fresh garlic (*Allium sativum* L.), Balady cultivar, was cleaned, manually peeled, and frozen at (~−40 ± 1 °C), then dried in a lyophilizer (Labconco 74200, Kansas, KS, USA) on 0.120 mbar, at 19 ± 1 °C and a condenser at −85 ± 1 °C. The garlic was ground and transferred to nanoparticles, as stated by Khataee, et al. [26]. The garlic was crushed with a grinder (Moulinex-Grinder; MC300, France) to fine garlic lyophilized powder (Ga-LP). Then, the particles were crushed by planetary ball mill (PM 2400, Karaj, Iran) (ball:powder ratio; 10:1) under a rotation velocity of 320 rpm for 2 h at ambient temperature (25 °C) to produce garlic nanoparticles. The garlic nanoparticles (Ga-NPs) were assessed with a Zetasizer (Nano-Sight NS300, UK) that was 90 ± 7 nm. The Ga-NPs were packaged in dark bottles until utilized according to Morsy, Morsy, Elbarbary and Saad [6]. Extraction of Ga-LP and Ga-NPs to obtain Ga-LPE and Ga-NPsE was done in an ultrasonic bath (Bandelin-Super Sonorex RK-100H), as asserted by Tabaraki, et al. [27]. The 50 mL methanol (70%) was added to a 1 g sample in a conical flask and sonicated for 30 min, then cooled to 25 ± 1 °C. The garlic extracts were passed through Whatman paper No. 1 and concentrated by rotary evaporator (IKA-WERKE; Germany) at a rotation speed of 250 rpm at 40 °C under vacuum. Samples were frozen at ~−40 ± 1 °C, then dried for 72 h in a lyophilizer. 

### 2.3. LC/MS/MS Fingerprint

The organosulfur compounds in garlic were determined using LC/MS/MS [28]. LC/MS/MS analyses were carried out by a Sciex 4000 QTRAP^®^ system LC system coupled a hybrid triple quadrupole LIT (linear ion trap), including Harvard apparatus 11 plus. Pump: The isolation was performed on a SUPELCO Discovery HS-F5 column (3 μm, 3 × 150 mm; Sigma- Aldrich Co., Steinheim Germany). The mobile phase contained A (water + 0.1% formic acid) and B (acetonitrile). The mobile phase carried out was: 0–15 min, B 0–100%; 15–25 min, B 100 %. The flow speed was 30 μL min^−1^, and the column oven was kept operational at 30 ± 1 °C. The injection volume was 5 μL. Software: Analyst^®^ 1.6.3 with Hotfix. 

### 2.4. Total Phenolics Content (TPC), Total Flavonoids Content (TFC), and Antioxidant Ability of Extracts

TPC was calculated using the Folin–Ciocalteu reagent, as described by Singleton et al. [29]. The results were expressed as mg GAE (gallic acid equivalent) per g^−1^ dw extract. Total flavonoids content (TFC) was determined in accordance with Zhishen, et al. [30] and expressed as mg RE (Rutin equivalents) per g^−1^ dw extract. The antioxidant ability of garlic extracts in terms of free radical scavenging potential was tested using DPPH assay. The outcomes were computed as IC_50_ (µg mL^−1^) [31]. The ABTS method was also used in accordance with Re, et al. [32], which measured at a wavelength of 734 nm.

### 2.5. Sunflower Oil (SFO) and Ga-NPs

The SFO (without synthetic antioxidants) was divided into seven groups. Garlic nanoparticles (Ga-NPs) were added to SFO in 3 groups at different concentrations, namely, 200 ppm, 600 ppm, and 1000 ppm (*w*/*v*). The 4th group contained Ga-LPE 600 ppm (*w*/*v*). The 5th and 6th groups included BHT and α-tocopherol at 200 ppm (*w*/*v*). The final one was SFO without antioxidant (control group). The oil samples were packed into dark glass bottles and stored under accelerated oxidation conditions (65 °C/24 days). Oil samples were collected at regular intervals for analysis at 0, 4, 8, 12, 16, 20, and 24 days of storage.

### 2.6. Oxidative Stability Parameters of SFO during Storage 

#### 2.6.1. Peroxide Value (PV) 

The peroxide value (PV) of SFO samples was performed as stated by Zhang, et al. [33]. The PV was calculated as meq kg^−1^ oil (milliequivalent per kilogram).

#### 2.6.2. p-Anisidine Value (*p*-AnV) and Totox Value (TV)

*p*-Anisidine value was carried out according to Chong, et al. [34] using the spectrophotometer (Model CM-5; Konica Minolta Sensing, Inc., Osaka, Japan) at 350 nm. The outcomes of *p*-AnV were calculated as mg kg^−1^ in accordance with the equation (Equation (1)):AV = 25 × [(1.2 As − Ab)]/W(1)
where As is the absorbance of oil sample with the *p*-anisidine reagent, Ab is the absorbance of blank, and W is the sample weight (g). Totox value (TV) was estimated using AOCS [35] using the Equation (2): Totox value = *p*-AnV + 2 PV(2)
where *p*-AnV is the *p*-anisidine value and PV is the peroxide value

#### 2.6.3. Conjugated Dienes (CDs) and Conjugated Trienes (CTs)

Determination of CDs and CTs of oil samples were performed using a spectrophotometer at a wavelength of 233 and 268 nm, respectively [36].

#### 2.6.4. Induction Period (IP), Antioxidant Efficiency, and Shelf Life Prediction

The induction period (IP) of the oil samples was estimated with a Metrohm 679 Rancimat instrument (Switzerland) at 110 ± 1 °C at a flow velocity of 15 Lh^−1^ [37]. Protection factor (PF) and antioxidant activity (AA) values of SFO were determined according to Bandonien, et al. [38] based on the following calculations (Equations (3) and (4)):PF = IP_S_/IP_C_(3)
AA = IP_S_ − IP_C_/IP_BHT_ − IP_C_(4)
where IP_S_ is the induction period of the sample (SFO) with an antioxidant additive, IP_c_ is the induction period of the control sample without an antioxidant additive, and IP_BHT_ is the induction period of the sample with BHT. 

The line slopes were measured by the plotting concentration, and time was used to calculate the temperature acceleration factor (Q10), which was based on the increase in oxidation level for every 10 °C increase in temperature, in accordance with [39] using the Equation (5): Q _10_= *e*
^(T1 − T2)/10^(5)
where *e* is the constant equal (2), T1 is the accelerated temperature (65 °C), and the T2 is ambient temperature (25 °C).

### 2.7. Sensory Evaluation

The organoleptic test of sunflower oil (SFO) was performed by an experienced and trained 12-member panel from the Biochemistry Department, according to Sadeghi, et al. [40]. The SFO treatments were placed into small dark glasses (~30 mL) with random numbers. All SFO samples were scored by the panel members using a hedonic (7-point) scale for color, aroma, and acceptability.

### 2.8. Statistical Analysis

The statistical significance in the SFO analysis was accomplished using the ANOVA test (SPSS 19, SPSS Inc., Chicago, IL, USA) followed by Tukey’s multiple comparison test, with a significance level of *p* < 0.05 [41]

## 3. Results and Discussion

### 3.1. Characterization of Garlic and Garlic Nanoparticles (Ga-NPs)

#### 3.1.1. Antioxidant Capacity

TPC and TFC are important constituents that can be used as an indicator of antioxidant capacities. Different cultivars of garlic such as Egyptian garlic (Balady), Italian Red, Sids-40, Chinese, and Brazilian Hozan samples were used. The TPCs in varied cultivars of garlic are presented in Table 1. The average TPCs ranged from 22.7–27.2 mg GAE g^−1^ extract (on dw). The TPC was highest in the Balady garlic cloves (*p* < 0.05) when compared to other cultivars. Petropoulos, Fernandes, Barros, Ciric, Sokovic and Ferreira [16] found that TPC values in garlic ranged from 8.59–44.85 mg GAE g^−1^ dw extract, while Kim, et al. [42] reported that the TPC in aged black garlic was 22.17 mg GAE g^−1^ dw. This variation could be attributed to cultivars or extraction methods. Total flavonoid content (TFC) in garlic cultivars ranged from 0.65–1.52 mg g^−1^ dw extract. These findings are consistent with those reported by Soto, et al. [43]. On the other hand, the nanoparticles of Balady garlic (Ga-NPs) had the highest TPC of 38.1 and TFC of 3 mg GAE g^−1^ compared to garlic (bulk shape). According to previous research, phenolics and flavonoids play an important role in antioxidant capacity [44]. 

Table 1 illustrates that the antioxidant activity (DPPH and ABTS) of Balady garlic extract and Ga-NPs. The data reveal that garlic extract has a high antioxidant ability of 11.8 and 10.8 μg mL^−1^ for DPPH and ABTS, respectively. The antioxidant ability of Balady Ga-NPsE was lower than BHT and higher than α-tocopherol. The results are consistent with data obtained by Iqbal and Bhanger [45]. The analysis of the data reveals that the antioxidant activity of garlic cultivars was correlated with TPC and TFC (R = 1). The TFC and TPC became increasingly important to antioxidant activity. [44,45] also found that antioxidant ability was positively correlated with TFC and TPC. 

#### 3.1.2. LC/MS/MS Fingerprint

The organosulfur compounds in garlic were determined using QTRAP LC/MS/MS analysis. The results in Figure 1 confirmed that garlic extracts have a significant number of organosulfur compounds. The LC/MS/MS recognized the major sulfur compounds (~4 peaks) in garlic extracts as alliin, allicin, S-allyl-mercapto-cysteine, and Glutamyl-(s)-Allyl-Cyste. The compounds’ precursor–product ion pairs (quantification transitions) were discovered to be *m*/*z* 178.1–88, 163.1–73.2, 194.0–111.2, and 291.3–145.2, respectively. These fragments were also observed by Zhu, Kakino, Nogami, Ohnuki and Shimizu [28], who found that garlic extract have alliin, S-allyl-L-cysteine, γ-glutamyl-S-allyl-L-cysteine, and allicin. These compounds are primarily responsible for the biological activities of garlic extract, which include antioxidant, antibacterial, and anticancer properties [46]. Sulfur compounds’ antioxidant activity is linked to their ability to reduce reactive oxygen species and activate antioxidative enzymes [47]. When exposed to negative conditions such as high temperatures, oxygen, and light, the organosulfur compounds found in garlic become volatile, thermally unstable, and prone to degradation and oxidation [48]. The characterization of bioactive compounds is dependent on the stability of garlic. Alliin is unstable and non-volatile due to the alliinase-catalyzed conversion into allicin when garlic is crushed. Alliinase reacts with alliin to produce reactive intermediates that combine to form allicin, an unstable alkenyl alkene thiosulfinate [49]. While allicin decomposes quickly in vitro to form a variety of organosulfur compounds, such as diallyl sulfides. This degradation was observed within hours at room temperature or during cooking [50].

### 3.2. Impact of Ga-NPsE on the Oxidative Stability of SFO Samples 

#### 3.2.1. Peroxide Value (PV)

Table 2 shows the addition of antioxidants and time of storage on PV had a significant impact on PV (*p* ≤ 0.05). In general, the PV increased for all samples with the progression of time. The PV increased in the following sequence: control > α-tocopherol > Ga-LPE 600 ppm > Ga-NPsE 200 ppm > Ga-NPsE 600 ppm > Ga-NPsE 1000 ppm > BHT meq kg^−1^. Sunflower oil treatments without the antioxidant (control) achieved the highest PV of 166.1 meq kg^−1^ after 24 days of storage at 65 °C. The increase in PV caused by lipid hydroperoxide formation may also degrade into volatile and non-volatile compounds, causing the oil’s quality to deteriorate [51]. PV was found to be significantly different between the control and garlic extract samples. The rate of peroxide formation was slowed at all concentrations where the PV recorded was 70.6, 46.4, 41.7, 38.6, 20.14, 20.2 meq kg^−1^ for α-tocopherol, Ga-LPE 600 ppm, Ga-NPsE 200 ppm, Ga-NPsE 600 ppm, Ga-NPsE 1000 ppm, and BHT, respectively, while the limiting value of peroxide was less than 21 meq kg^−1^ oil. Samples containing α-tocopherol, on the other hand, had significantly higher values than all Ga-NPsE and Ga-LPE samples combined. However, the PV in BHT-containing SFO samples was lower than in garlic-containing samples [45].

#### 3.2.2. *p*-Anisidine Value (p-AnV)

The *p*-anisidine value (*p*-AnV) is a measurement of the secondary fat oxidation output such as aliphatic aldehydes, ketones, alcohols, acids, and hydrocarbons. In this case, the edible oils develop off flavors and off odors. Table 2 shows the results, which reveal increases in *p*-AnV for all the SFO samples. The *p*-AnV concentration in the control group significantly increased from 4.20 to 25.26 mg kg^−1^ until day 24 of storage. Ga-NPsE (1000 ppm) and BHT-incorporated oil showed the lowest amount of *p*-AnV compared to α-tocopherol, Ga-LPE 600 ppm, Ga-NPsE 200 ppm, and Ga-NPsE 600 ppm. The threshold of *p*-AnV in oil is less than 10 mg kg^−1^. These findings are consistent with the findings reported by Nyam, et al. [52], who found that the natural extracts have a notable reduction impact against the oxidation of SFO under accelerated storage. Although Ga-NP extract was significantly lower in the *p*-AnV, the efficiency of BHT synthetic antioxidants was slightly higher than Ga-NPsE.

#### 3.2.3. Totox Value (TV)

Totox value (TV) indicates the presence of primary and secondary oxidation compounds, indicating the early and later stages of oxidative rancidity. As a result, it provides a more accurate estimate of oil deterioration and quality. There was a noticeable difference in the totox value of the treated SFO and the control (Table 3). After 24 days of storage, the maximum values were 357.5 and 160.7 meq kg^−1^ for the control samples and α-tocopherol-enriched sample, respectively, while those of oil samples enriched with BHT and Ga-NPs 1000 ppm Ga-NPsE had the lowest TV.

#### 3.2.4. Conjugated Dienes (CDs) and Conjugated Trienes (CTs)

Table 4 shows the formation of CDs and CTs in the control and stabilized SFO treatments during storage. Generally, the CD values of all groups increased. The highest contents were observed for the control sample, while the addition of garlic reduced the number of CDs. BHT recorded the lowest value (6.15). Ga-NPsE (1000 ppm) had a low value compared to the samples containing non-synthetic antioxidants. Furthermore, the values of CTs were increased significantly with storage time in all treatments (Table 4). CT values were lowest in oil samples supplemented with BHT and Ga-NPs 1000. The CDs and CTs of the SFO samples supplemented with BHT were significantly lower than those given garlic extract-treated samples [52].

### 3.3. Induction Period (IP) and Antioxidant Efficiency of Ga-NPsE 

Table 5 demonstrates that all SFO samples were significantly different (*p* ≤ 0.05) in IP values than that of the control. IP values in samples containing Ga-NPs were 15.4, 18.7, and 22.2 h, for Ga-NPs concentrations of 200 ppm, 600 ppm, and 1000 ppm, respectively. It was noted that there are significant differences in the samples’ IP scores, including different concentrations of Ga-NPs. The SFO sample containing BHT exhibited the highest IP (25.3 h). The findings agree with those reported by Carelli, et al. [53], who found that natural antioxidants increased the IP of SFO. Table 5 shows that the values of the protection factor (PF) of SFO samples including garlic extracts ranged from 1.21 to 3.33. The high concentrations of Ga-NPsE showed a significant increase in PF value. Moreover, the antioxidant activity (AA) of samples containing BHT and Ga-NPsE 1000 ppm did not differ significantly (*p* ≥ 0.05). In addition, we observed that AA increased up to 1 in SFO containing BHT. Generally, adding Ga-NPsE to SFO samples significantly increased the antioxidant efficiency (AE).

### 3.4. Impact of Ga-NPsE and/or Ga-LPE (600 ppm) on Stability 

Figure 2 illustrates the differences in the oxidation parameters (PV, *p*-AnV, and CD value) after 24 days of storage of SFO incorporating 600 ppm Ga-NPsE and/or Ga-LPE 600. The oxidation parameters of samples containing Ga-NPsE 600 ppm and Ga-LPE 600 ppm differed significantly (*p* ≤ 0.05). Moreover, the PV, *p*-AnV, and CD values of Ga-NPsE 600 ppm (25.1, 12.8, and 8.2, respectively) were lower than those of Ga-LPE (46.4, 17.7, and 13.5, respectively). The results confirm that Ga-NPsE enhanced the oxidative stability of SFO.

### 3.5. Sensory Evaluation of SFO Incorporating Ga-NPsE 

Figure 3 depicts the sensory parameters, such as color, aroma, and acceptability, of sunflower oil (SFO) treatments during storage. The sensory assessment of SFO samples was done for different storage periods caused by the ending of shelf life with the intelligible development of an off flavor. Panelists generally approved of all sensory properties of SFO treatments at time zero; as the storage progressed, the sensory acceptability decreased. The control sample scored the lowest color value during the storage periods, while the oil sample incorporating Ga-NPs at different levels were accepted up to day 24. The aroma value differed significantly (*p* ≤ 0.05) between the oil treatments and the control group. SFO samples incorporating BHT, α-tocopherol or Ga-NPS 1000 ppm were accepted until day 24 of storage; however, the control was rejected on day 12 due to an off flavor. SFO samples’ acceptability values decreased significantly over time, whereas BHT and Ga-NPS 1000 ppm samples did not differ after 24 days. The sensory data demonstrate that the control sample was rejected after day 12 of storage; however, the oil samples incorporating BHT, α-tocopherol, and Ga-NPS 1000 ppm were accepted up to day 24. These changes match well with chemical indicators, i.e., PV, p-AnV, and TV. These findings are consistent with those noted by Iqbal and Bhanger [45], who found that the chemical and sensory parameters are important to evaluate the efficacy of garlic antioxidants in sunflower oil.

The shelf life of SFO at 65 °C and predictable shelf life during storage at 25 °C are shown in (Figure 4). The data reveal that the shelf life of SFO at 65 °C was 12 days for the control sample, 16 days for Ga-NPsE 200 ppm, 20 days for Ga-NPsE 600 ppm, 24 days for Ga-NPsE 1000 ppm, 24 days for BHT, and 24 days for α-tocopherol. Nevertheless, the expected shelf life of SFO at 25 °C ranged from 192–384 days. Consequently, the acceleration factor was 16 days (Q_10_ value for fat oxidation of 2.0); this signifies that one day at 65 °C is equal to 16 days at 25 °C [54].

## 4. Conclusions

It can be concluded that the Ga-NPsE, Balady cultivar, may be used as an alternative antioxidant to the synthetic BHT against the oxidation reactions in SFO through storage at 65 ± 1 °C/24 days. A total of four organosulfur compounds in Ga-NPs were identified based on QTRAP LC/MS/MS. The highest phenolic, flavonoid, and antioxidant ability was found in Ga-NPsE. In Ga-NPsE-treated SFO, the values of peroxide, p-anisidine, totox, conjugated dienes, and conjugated trienes were significantly lower than in the control. The antioxidant indices of SFO samples containing Ga-NPs were higher than the control. The Ga-NPsE enhanced the sensory acceptability of SFO treatments up to 24 days of storage. The garlic nanoparticles are more active due to their small size, widespread distribution, and interaction with various antioxidant modes of action. The results show that Ga-NPsE is an effective antioxidant for enhancing the stability of SFO and extending its shelf life (~384 days at 25 °C). However, more experiments are required to test the efficacy of Ga-NPsE antioxidants such as biological-based antioxidant mechanisms (electron transfer or inhibition of lipid peroxidation). 

## Figures and Tables

**Figure 1 foods-11-03962-f001:**
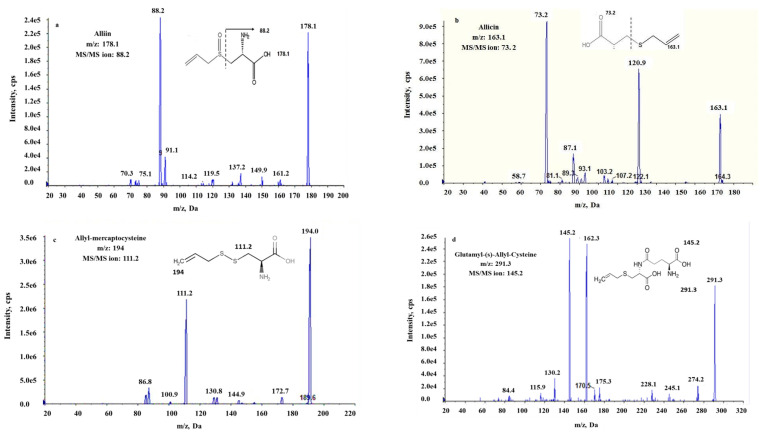
Product ion spectra and the main fragmentation pathway of [M + H]+ for alliin (**a**), allicin (**b**), allyl-mercapto-cysteine (**c**), and Glutamyl-(s)-Allyl-Cysteine (**d**).

**Figure 2 foods-11-03962-f002:**
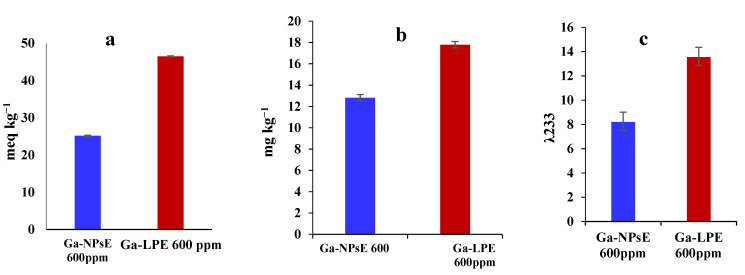
Effect of Ga-NPsE and Ga-LPE (600 ppm) on the oxidation parameters of SFO after 24 days of storage. Peroxide value (**a**), p-anisidine value (**b**), and conjugated dienes (**c**).

**Figure 3 foods-11-03962-f003:**
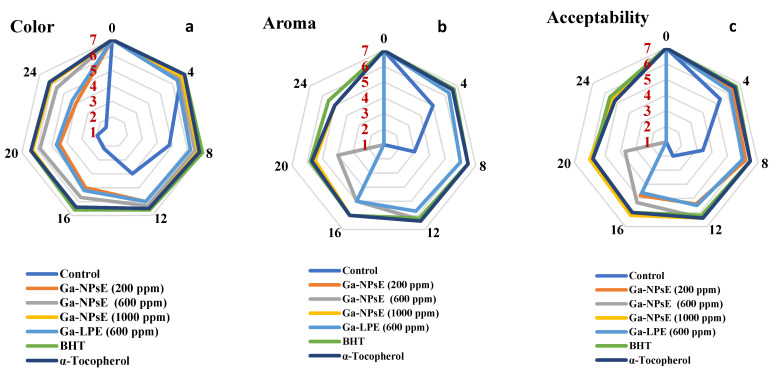
Impact of the addition of Ga-NPs on color (**a**), aroma (**b**), and acceptability (**c**) in SFO samples during storage at (65 °C).

**Figure 4 foods-11-03962-f004:**
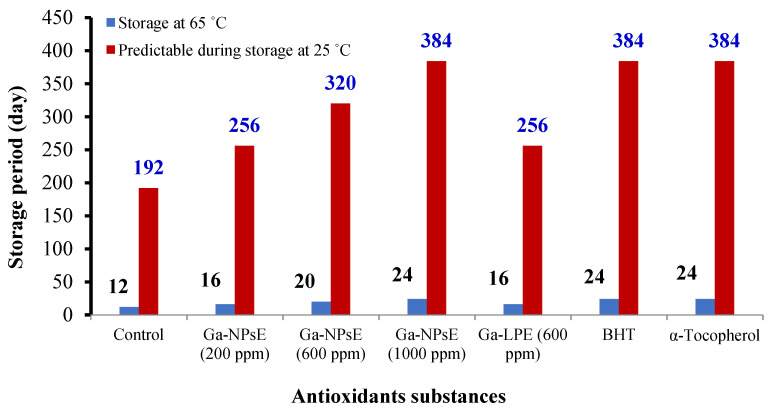
The shelf life of SFO during storage at 65 °C and predictable shelf life during storage at 25 °C.

**Table 1 foods-11-03962-t001:** Total phenolic content (TPC), Total flavonoid content (TFC), and antioxidant activities (AA) of garlic extracts from different cultivars compared to BHT and α-tocopherol (mean ± SD, *n* = 3).

Sample	TPC (mg g^−1^ dw Extract)	TFC (mg g^−1^ dw Extract)	DPPH IC_50_ (μg mL^−1^)	ABTS assay (μg mL^−1^)	R
Balady	27.25 ± 0.45 ^b^	1.52 ± 0.14 ^b^	11.87 ± 0.76 ^d^	10.88 ± 1.51 ^e^	1
Italian Red	25.35 ± 0.88 ^c^	0. 91 ± 0.65 ^c^	12.14 ± 0.41 ^d^	11.34 ± 0.87 ^d^	1
Sids-40	24.37 ± 1.07 ^c^	0.65 ± 0.33 ^d^	14.41 ± 0.32 ^c^	13.11 ± 0.23 ^c^	1
Chinese	23.30 ± 0.78 ^d^	0.74 ± 0.65 ^d^	17.77 ± 2.06 ^a^	15.55 ± 0.22 ^b^	1
Brazilian Hozan	22.76 ± 1.56 ^d^	0.72 ± 1.33 ^d^	18.30 ± 1.65 ^a^	17.17 ± 0.44 ^a^	1
BHT (200 ppm)	-	-	5.05 ± 0.87 ^f^	3.79 ± 1.42 ^g^	
α-Tocopherol (200 ppm)	-	-	16.15 ± 1.61 ^b^	13.21 ± 0.55 ^c^	

^a–g^: There are no significant differences between any two means in the same column that have the same lowercase superscript letter (*p* ≥ 0.05). TPC: Total phenolic content. TFC: Total flavonoid content. R: Pearson’s correlation coefficient.

**Table 2 foods-11-03962-t002:** Effect of garlic nanoparticle extracts on peroxide value and *p*-anisidine value of SFO during storage at 65 °C (mean ± SD, *n* = 3).

Sample	Storage Period (day)						
	Time Zero	4	8	12	16	20	24
	Peroxide value (meq kg^−1^)						
Control	1.20 ± 0.02 ^a G^	20.22 ± 0.15 ^a F^	25.15 ± 0.33 ^b E^	40.54 ± 0.21^a D^	78.00 ± 0.02 ^a C^	170.11 ± 0.56 ^a A^	166.16 ± 0.34 ^a B^
Ga-NPsE (200 ppm)	1.03 ± 0.01 ^a G^	17.39 ± 0.21^b F^	24.09 ± 0.04 ^b E^	31.16 ± 0.43 ^c D^	35.25 ± 0.6 ^d C^	39.67 ± 0.09 ^c B^	41.78 ± 0.29 ^b A^
Ga-NPsE (600 ppm)	1.01 ± 0.03 ^a G^	12.30 ± 0.02 ^c F^	14.43 ± 0.13 ^c E^	18.45 ± 0.32 ^d D^	28.13 ± 0.17 ^e C^	32.09 ± 0.22 ^d B^	38.65 ± 0.11^d A^
Ga-NPsE (1000 ppm)	1.03 ± 0.02 ^a G^	5.24 ± 0.04 ^a F^	10.16 ± 0.07 ^d E^	13.32 ± 0.01 ^e D^	17.75 ± 0.05 ^f C^	18.81 ± 0.46 ^f B^	20.14 ± 0.16 ^e A^
Ga-LPE (600 ppm)	1.12 ± 0.04 ^a G^	16.44 ± 0.06 ^b F^	28.54 ± 0.11^a E^	35.33 ± 0.05 ^b D^	37.87 ± 0.28 ^c C^	39.34 ± 0.03 ^c B^	46.43 ± 0.12 ^c A^
BHT (200 ppm)	1.11 ± 0.01 ^a G^	3.12 ± 0.16 ^d F^	4.16 ± 0.10 ^e E^	10.52 ± 0.14 ^f D^	13.66 ± 0.26 ^g C^	16.54 ± 0.12 ^g B^	20.22 ± 0.18 ^f A^
α-Tocopherol (200 ppm)	1.02 ± 0.01 ^a G^	15.14 ± 0.31^c F^	28.11 ± 0.07 ^a E^	40.43 ± 0.15 ^a D^	57.16 ± 0.18 ^b C^	65.36 ± 0.43 ^b B^	70.62 ± 0.07 ^b A^
	*p*-Anisidine value (mg kg^−1^)						
Control	4.20 ± 0.04 ^a G^	6.52 ± 0.25 ^a F^	11.65 ± 0.15 ^a E^	14.38 ± 0.31^a D^	16.55 ± 0.42 ^a C^	19.12 ± 0.11 ^a B^	25.26 ± 0.04 ^a A^
Ga-NPsE (200 ppm)	4.12 ± 0.10 ^a G^	5.04 ± 0.12 ^b F^	7.31 ± 0.30 ^c E^	9.96 ± 0.05 ^d D^	11.44 ± 0.05 ^c C^	14.22 ± 0.12 ^d B^	16.65 ± 0.02 ^d A^
Ga-NPsE (600 ppm)	4.03 ± 0.03 ^a G^	4.80 ± 0.22 ^c F^	6.93 ± 0.09 ^c E^	7.46 ± 0.82 ^e D^	9.12 ± 0.21 ^d C^	10.12 ± 0.06 ^e B^	12.81 ± 0.31 ^e A^
Ga-NPsE (1000 ppm)	4.11 ± 0.23 ^a G^	4.24 ± 0.04 ^c F^	5.19 ± 0.11 ^d E^	6.15 ± 0.04 ^f D^	7.75 ± 0.23 ^e C^	8.83 ± 0.04 ^f B^	10.02 ± 0.12 ^f A^
Ga-LPE (600 ppm)	4.31 ± 0.05 ^a G^	5.49 ± 0.20 ^b F^	7.75 ± 0.57 ^c E^	12.04 ± 0.03 ^c D^	15.27 ± 0.06 ^b C^	16.11 ± 0.17 ^c B^	17.78 ± 0.28 ^c A^
BHT (200 ppm)	3.98 ± 0.21 ^a G^	4.33 ± 0.01 ^c F^	4.19 ± 0.07 ^e E^	5.02 ± 0.13 ^g D^	6.68 ± 0.09 ^f C^	8.84 ± 0.12 ^f B^	9.72 ± 0.09 ^g A^
α-Tocopherol (200 ppm)	4.02 ± 0.09 ^a G^	5.56 ± 0.21 ^b F^	8.67± 0.02 ^b E^	13.43 ± 0.12 ^b D^	16.91 ± 0.01 ^a C^	17.93 ± 0.30 ^b B^	19.54 ± 0.23 ^b A^

^a–g^: There are no significant differences between any two means in the same column that have the same lowercase superscript letter (*p* ≥ 0.05). ^A–G^: There are no significant differences between any two means in the same row that have the same uppercase superscript letter (*p* ≥ 0.05). Ga-NPsE: garlic nanoparticles extract; Ga-LPE: garlic lyophilized powder extract.

**Table 3 foods-11-03962-t003:** Effect of garlic nanoparticle extracts on the totox value (meq kg^−1^) of SFO during storage at 65 °C (mean ± SD, *n* = 3).

Sample	Storage Period (day)						
	Time Zero	4	8	12	16	20	24
Control	6.61 ± 0.31 ^a G^	46.97 ± 0.71 ^a F^	61.95 ± 0.73 ^b E^	95.47 ± 0.29 ^a D^	172.55 ± 0.98 ^a C^	359.33 ± 0.67 ^a A^	357.58 ± 1.17 ^a B^
Ga-NPsE (200 ppm)	6.36 ± 0.55 ^a G^	40.26 ± 0.82 ^b F^	55.94± 0.08 ^c E^	74.36 ± 0.23 ^c D^	85.77 ± 0.75 ^d C^	95.46 ± 0.53 ^c B^	101.34 ± 0.7 ^d A^
Ga-NPsE (600 ppm)	6.05 ± 0.40 ^a G^	29.4 ± 0.45 ^e F^	35.79 ± 0.51 ^d E^	44.36 ± 1.2 ^d D^	65.37 ± 0.69 ^e C^	74.31 ± 0.54 ^e B^	90.11 ± 1.3 ^e A^
Ga-NPsE (1000 ppm)	6.16 ± 0.24 ^a G^	14.72 ± 0.39 ^f F^	25.51 ± 0.9 ^e E^	32.8 ± 0.60 ^e D^	43.25 ± 0.19 ^f C^	52.45 ± 0.34 ^f B^	60.31 ± 1.1^f A^
Ga-LPE (600 ppm)	6.35 ± 0.32 ^a G^	37.91 ± 0.37 ^c F^	64.39± 0.33 ^a E^	80.61 ± 0.96 ^b D^	87.18 ± 0.39 ^c C^	92.9 ± 0.36 ^d B^	109.51 ± 0.98 ^c A^
BHT (200 ppm)	6.21 ± 0.29 ^a G^	10.58 ± 0.93 ^g F^	12.51 ± 0.11 ^f E^	26.06 ± 0.41^f D^	34 ± 0.92 ^g C^	41.92 ± 0.43 ^g B^	50.15 ± 0.87 ^g A^
α-Tocopherol (200 ppm)	6.06 ± 0.62 ^a G^	35.85 ± 0.81^d F^	64.88± 0.97 ^a E^	94.29 ± 0.12 ^a D^	131.23 ± 0.88 ^b C^	148.66 ± 0.77 ^b B^	160.78 ± 0.96 ^b A^

^a–g^: There are no significant differences between any two means in the same column that have the same lowercase superscript letter (*p* ≥ 0.05). ^A–G^: There are no significant differences between any two means in the same row that have the same uppercase superscript letter (*p* ≥ 0.05). Ga-NPsE: garlic nanoparticles extract; Ga-LPE: garlic lyophilized powder extract. These results could be attributed to bioactive compounds of garlic such as allicin, 1,2-vinyldithiin, allicin, and S-allyl-cysteine [46].

**Table 4 foods-11-03962-t004:** Effect of garlic nanoparticle extracts on conjugated dienes and conjugated trienes of SFO during storage at 65 °C (mean ± SD, *n* = 3).

Sample	Storage Period (day)						
	Time Zero	4	8	12	16	20	24
	Conjugated dienes value (ε^1%^_1cm_ λ_233_)						
Control	2.50 ± 0.03 ^a G^	6.17 ± 0.04 ^a F^	9.19 ± 0.10 ^a E^	11.07 ± 0.2 ^a D^	13.65 ± 0.11 ^a C^	15.22 ± 0.67 ^a B^	19.22 ± 0.01 ^a A^
Ga-NPsE (200 ppm)	2.46 ± 0.06 ^a G^	4.11 ± 0.2 ^b F^	5.34± 0.06 ^c E^	7.01 ± 0.9 ^d D^	8.08 ± 0.03 ^d C^	9.00 ± 0.33 ^d B^	12.11± 0.98 ^d A^
Ga-NPsE (600 ppm)	2.45 ± 0.04 ^a G^	4.04 ± 0.04 ^b F^	5.00 ± 0.05 ^c E^	5.06± 0.02 ^e D^	5.37 ± 0.06 ^e C^	6.13 ± 0.51 ^e B^	8.21 ± 0.8 ^e A^
Ga-NPsE (1000 ppm)	2.45 ± 0.09^a G^	3.15 ± 0.03 ^c F^	3.50 ± 0.1^d E^	4.10 ± 0.40 ^f D^	4.65 ± 0.19 ^f C^	5.05 ± 0.3 ^f B^	6.71 ± 0.01 ^f A^
Ga-LPE (600 ppm)	2.47 ± 0.09 ^a G^	4.33 ± 0.82 ^b F^	5.82± 0.08 ^c E^	8.18 ± 0.02 ^c D^	9.79 ±0.06 ^c C^	10.14 ± 0.05 ^c B^	13.55 ± 0.7 ^c A^
BHT (200 ppm)	2.41 ± 0.07 ^a G^	2.98 ± 0.14 ^d F^	3.53 ± 0.21 ^d E^	3.91 ± 0.04 ^g D^	4.61 ± 0.12 ^f C^	5.0 ± 0.21 ^f B^	6.15 ± 0.07 ^f A^
α-Tocopherol (200 ppm)	2.46 ± 0.07 ^a G^	4.29 ± 0.17 ^b F^	7.46± 0.27 ^b E^	9.21 ± 0.22 ^b D^	10.00 ± 0.23 ^b C^	12.56 ± 0.30 ^b B^	16.00 ± 0.09 ^b A^
	Conjugated trienes value (ε^1%^_1cm_ λ_268_)						
Control	0.54 ± 0.03 ^a G^	1.55 ± 0.01 ^a F^	1.89 ± 0.10 ^a E^	2.09 ± 0.02 ^a D^	2.60 ± 0.01 ^a C^	3.00 ± 0.01 ^a B^	4.12 ± 0.02 ^a A^
Ga-NPsE (200 ppm)	0.53± 0.04 ^a G^	1.00 ± 0.2 ^a F^	1.22± 0.04 ^c E^	1.61 ± 0.9 ^b D^	1.78 ± 0.03 ^b C^	2.54 ± 0.03 ^b B^	3.42 ± 0.09 ^b A^
Ga-NPsE (600 ppm)	0.52 ± 0.03 ^a G^	0.94 ± 0.03 ^b F^	1.01 ± 0.05 ^c E^	1.39± 0.1^b D^	1.62 ± 0.04 ^c C^	2.22 ± 0.02 ^c B^	3.33 ± 0.01^b A^
Ga-NPsE (1000 ppm)	0.52 ± 0.03^a G^	0.81 ± 0.01^b F^	0.91 ± 0.01^d E^	1.20 ± 0.04 ^b D^	1.42 ± 0.01 ^c C^	2.15 ± 0.03 ^c B^	3.11 ± 0.07 ^c A^
Ga-LPE (600 ppm)	0.53 ± 0.07 ^a G^	1.12 ± 0.02^a F^	1.14± 0.03 ^c E^	1.88 ± 0.02 ^a D^	2.15 ±0.03 ^b C^	2.73 ± 0.02 ^b B^	3.54 ± 0.05 ^b A^
BHT (200 ppm)	0.51 ± 0.06 ^a G^	0.62 ± 0.01 ^c F^	0.87 ± 0.02 ^d E^	1.11 ± 0.04 ^c D^	1.33 ± 0.01^c C^	1.98 ± 0.02 ^d B^	3.00 ± 0.05 ^d A^
α-Tocopherol (200 ppm)	0.53 ± 0.09 ^a G^	1.23 ± 0.01^b F^	1.43± 0.07 ^b E^	1.98 ± 0.04 ^a D^	2.26 ± 0.01 ^b C^	2.86 ± 0.03 ^a B^	3.7 ± 0.17 ^b A^

^a–d^: There are no significant differences between any two means in the same column that have the same lowercase superscript letter (*p* ≥ 0.05). ^A–G^: There are no significant differences between any two means in the same row that have the same uppercase superscript letter (*p* ≥ 0.05). Ga-NPsE: garlic nanoparticles extract; Ga-LPE: garlic lyophilized powder extract.

**Table 5 foods-11-03962-t005:** Induction period (IP), protection factor (PF), and antioxidant activity (AA) of garlic nanoparticle extracts during storage at 65 °C (mean ± SD, *n* = 3).

	IP (h)	PF	AA
Control	7.6 ± 0.11 ^g^	-	-
Ga-NPsE (200 ppm)	15.4 ± 0.15^d^	2.03± 0.12 ^d^	0.44 ± 0.09 ^c^
Ga-NPsE (600 ppm)	18.7 ± 0.30 ^c^	2.46± 0.31 ^c^	0.63 ± 0.1^b^
Ga-NPsE (1000 ppm)	22.2 ± 0.24 ^b^	2.92± 0.15 ^b^	0.82 ± 0.08 ^a^
Ga-LPE (600 ppm)	12.0 ± 0.2 ^e^	1.57± 0.03 ^e^	0.27 ± 0.21 ^d^
BHT (200 ppm)	25.3 ± 0.21 ^a^	3.33± 0.09 ^a^	1 ± 0.11^a^
α-Tocopherol (200 ppm)	9.2 ± 0.32^f^	1.21 ± 0.15 ^f^	0.9 ± 0.12 ^a^

^a–g^: There are no significant differences between any two means in the same column that have the same lowercase superscript letter (*p* ≥ 0.05). Ga-NPsE: garlic nanoparticles extract; Ga-LPE: garlic lyophilized powder extract. IP: induction period, PF: protection factor, AA: antioxidant activity.

## Data Availability

Available upon request from the corresponding author.

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
