# Peer review of "QTRAP LC/MS/MS of Garlic Nanoparticles and Improving Sunflower Oil Stabilization during Accelerated Shelf Life Storage"

_foods, 2022, doi:10.3390/foods11243962_

Round 1
Reviewer 1 Report
This paper reports an extensive evaluation on the ability of garlic extract nanoparticles to lower the oxidative degradation of sunflower oils.
The paper is generally quite robust and provides some convincign data in terms of the improved effects of the garlic nanoparticle formulations/additives
I feel ther paper will be of relevance and interest to the journal of medicinal food readership subject to the corrections/ammendments listed below:-
1. mass spectra in FIg 1b: the chemical structure shown is not allicin (please correct)
2. Section 3.1.2: the authors state that allin is unstable due to its thiol group. This is incorrect. Firstly alliin doe not contain a thiol group. the instability of alliin in not due to the molecules structure but due to the elliinase catalysed conversion into allicin when garlic is crushed (this need to be clarified/corrected in this section.
3. in table-1 stability of extracts from several garlic cultivars are compared with those of a garlic nanoparticle preparation. which garlci cultivar were theh nanoparticles prepared from (this needs to be clarified in the table, the discussion and in the experimental procedures). This will then allow the authors to also include more meaningful discussion and comparison between the nanoparticle preparation and the standard extract preparation from the same cultiver.
4. section 3.1.2, line 9 spelling of "compounds"
5. In the conclusions, the paper would be greatly strengthened and more meaningful to the readers if some more insight could be offered into what reasons could be for the mproved performance of the nanonparticle preparations.
6. In the introduction it would be useful if some explanation could be included about why garlic nanopartile preparations were chosen for this study (for example, what is it abotu nanoparticle preparations that supported the initial hypothesis that they could potentially improve the oxidative stability of sunflower oils
Author Response
To Whom It May Concern;
Enclosed is the revised manuscript entitled “QTRAP LC/MS/MS of Garlic Nanoparticles and Enhancing Sunflower Oil's Oxidative Stability under Accelerated Storage”
We thank the Scientific Editor and all the reviewers for their comments and suggestions, as they have significantly improved the manuscript. The following are responses to the reviewers' comments. We hope that the manuscript is acceptable after these revisions.
Reviewers' comments
Reviewer # 1
- Mass spectra in Fig 1b: the chemical structure shown is not allicin (please correct).
We appreciate the time spent on reviewing our manuscript as well as your criticisms and critiques of the reviewed manuscript. I agree with your opinion, the correction was done in the manuscript.
- Section 3.1.2: the authors state that allin is unstable due to its thiol group. This is incorrect. Firstly alliin does not contain a thiol group. The instability of alliin is not due to the molecule's structure but due to the elliinase-catalyzed conversion into allicin when garlic is crushed (this need to be clarified/corrected in this section.
Ok, all corrections were done in the manuscript
- In the table-1 stability of extracts from several garlic, cultivars are compared with those of a garlic nanoparticle preparation. Which garlic cultivar were the nanoparticles prepared from (this needs to be clarified in the table, the discussion, and in the experimental procedures). This will then allow the authors to also include more meaningful discussion and comparison between the nanoparticle preparation and the standard extract preparation from the same cultivar.
The variety used for producing nanoparticles is Balady, the correction done in the manuscript.
- Section 3.1.2, line 9 spelling of "compounds"
Ok, done in the manuscript
- In the conclusions, the paper would be greatly strengthened and more meaningful to the readers if some more insight could be offered into what reasons could be for the improved performance of the nanoparticle preparations.
Ok, I agree with your opinion. The reasons were done in the manuscript
- In the introduction, it would be useful if some explanation could be included about why garlic nanoparticle preparations were chosen for this study (for example, what is it about nanoparticle preparations that supported the initial hypothesis that they could potentially improve the oxidative stability of sunflower oils.
Already we mentioned the previous studies used the nanoparticle from different plants to improve the oxidative stability of the oil. For example, One study by Morsy, Morsy, Elbarbary, and Saad [6] found that spirulina nanoparticles slowed lipid oxidation in olive oil during storage. Another investigation by Shehzad, et al. [21] found that nano-encapsulation-based curcumin has antioxidant activity in fish oil. Previous research attempted to improve the storage stability of sunflower oil by using natural antioxidants such as potato peels [22], grape seed [23], rosemary extract [24], and thyme extract [25].
Again, thank you for the opportunity to submit our manuscript to Foods Journal. If I can be of additional assistance with regard to this submission, please get in touch with me directly at +2 0122 3831825 or via email at mohamed.abdelhafez@fagr.bu.edu.eg
Regards,
Mohamed K Morsy, Ph. D.
Associate Professor at Food Technology Dept.

Reviewer 2 Report
The manuscript needs to be selectively improved regarding the language and writing style, quality of images and other relevant questions raised and commented on as follows.
The lack of line numbering difficulties in the review.
Abstract. Please correct the usage of the degree Celsius throughout the abstract and other parts of the manuscript.
Keywords. Please replace those from the title.
BHT and TBHQ are safe to use in foods when added in the proper amount. The reported risks to health are mainly related to higher doses than that indicated by surveillance agencies. Please reword your sentence in order not to mislead the readers.
This sentence has no sense: “Thence, the application of natural antioxidants extracted from plants and/or alga has received a lot of attention recently [6], essential oil [7], and sesame peel [8].”
GRAS and other acronyms must be introduced in their first mention.
Equations must be appropriately organized; some of them are totally disorganized and not adequately edited.
Please standardize the use of SI units according to the most reported, e.g., “h” instead of “hr”; sometimes, the authors use (p≥0.05), sometimes (P≥0.05).
Section 2.6.4 – Why do the authors performed Rancimat analysis at 100 °C when the AOCS indicates 110°C? Besides, the reference cited (which is outdated) indicates using three temperatures (100-120 °C), so anyone can predict the oil’s shelf life.
Furthermore, shelf-life prediction based on Ref [39] using the Q10 coefficient must be adequately explained; it is not possible to understand how the authors calculated it using the induction period from a single temperature measurement.
The text must be proofread for grammar, as there are some mistakes and issues. For instance, “The organoleptic test of sunflower oil (SFO) were done”
In Section 3.1.1, if the authors mention all the mean values from the Tables, they must choose to present only some ranges in the text or eliminate the table. Presenting both is not reasonable.
Other sentences with no grammar sense: “The data revealed that garlic extract has a high antioxidant ability was 11.8 and 10.8 μg mL-1 for DPPH and ABTS, respectively.” “Results illustrated in Fig 1 confirm that garlic extracts a significant amount of organosulfur compounds.” Please proofread the manuscript.
The authors say that “From the abovementioned results, Ga-NPs could be employed as an antioxidant substitute in oils, fats, and fatty foods,” but I disagree. There is no how to reach such a conclusion using only two antioxidant potential methods mainly based on radical scavenging. Other chemical and biological-based antioxidant mechanisms, such as electron transfer or inhibition of lipid peroxidation, must be accomplished to say that properly. The tests performed support antiradical activity only.
Footnotes from Tables must be corrected to indicate clearly that lowercase/uppercase letters indicate significant differences.
Tables 2, 3 and 4 could be merged into one, as their data all represent quality indicators. There is no mention of what the R2 represents. Also, there is no reasonability of an R2 equal to 1 (Ga-NPsE (1000 ppm)). In such cases, the authors may increase decimal digits. The authors did not standardize the number of significant digits in some tables. Include the concentration of BHT and a-tocopherol in the tables.
Clearly, Figure 1 has been produced based on the print screen, as some non-recognized words have been marked. The authors must edit properly and provide better-quality images. The resolution is too low.
In section 3.2.1, the authors commented on peroxide values, but there is no mention of the standards of such parameters for oils. Are the samples within the standard limits recommended by the regulatory agencies? Also, what do the authors mean with “GaNPsE 1000 ppm had R2=1, after 24 days of storage”?
The authors state that the p-anisidine value “slightly increased from 4.20 to 25.26 mg kg-1 until day 24 of storage”, but such an increase is not slight; it is significantly high. The following conclusion stated by the authors is also controversial, as the data indicates that the antioxidants were able to reduce but not avoid the increase in the p-AV. Be careful with such conclusions so as not to mislead the readers; the Tables disagree with the discussion in this regard.
Another sentence with no sense: “Moreover, R2 = 1 for Ga-NPsE (1000 ppm) and 0.89 for control.”
In the following sentence, “The findings agree with those reported by Carelli et al. [53],” what those authors concluded that your findings agree with? Please be specific, as the manuscript is full of such sentences without proper contextualization.
Another sentence with no sense: “No significant differences in antioxidant ability (AA) scores between samples containing GaNPsE 1000 ppm and BHT (P≥0.05).”
Charts from Figure 2 must be appropriately placed side-by-side for better visualization. In fact, they seem to represent the same data from the Tables. If so, there is no need for such figures. Please double-check.
In the same way, charts from Figure 3 must be appropriately placed side-by-side for better visualization and to avoid splitting them into different pages.
There is no suitable discussion and contextualization of the sensory scores,
This sentence brings all the information from Figure 4 and should be revised, or Figure 4 removed: “Nevertheless, the expected shelf life of SFO at 25 °C was ∼192, 256, 320, 384, 256, 384, and 384 days, respectively.”
Conclusion section. The authors cannot infer such a conclusion that Ga-NPsE can be used instead of BHT with such preliminary data based on 24 days of storage at a single temperature; many more tests must be performed to reach such an exact conclusion. They could merely suggest that such Ga-NPsE may be an alternative antioxidant but indicate that more tests must be assessed to prove such behavior better.
The following sentence is not a conclusion, but material and methods: “Ga-NPsE were placed into SFO at three levels i.e. 200, 600, and 1000 ppm (w/v), and compared with 600 ppm garlic lyophilized powder extract (Ga-LPE), 200 ppm BHT, and 200 ppm α-tocopherol and without Ga-NPsE (control).”
General comment:
Nowadays, the authors must be aware that the first step in the publication process is to provide a high-quality document to the reviewers, so we can focus on suggesting improvements to the main content of the study, other than the language, writing style, figures, tables, and format.
Author Response
To Whom It May Concern;
Enclosed is the revised manuscript entitled “QTRAP LC/MS/MS of Garlic Nanoparticles and Enhancing Sunflower Oil's Oxidative Stability under Accelerated Storage”
We thank the Scientific Editor and all the reviewers for their comments and suggestions, as they have significantly improved the manuscript. The following are responses to the reviewers' comments. We hope that the manuscript is acceptable after these revisions.
Reviewer # 2
- The lack of line numbering difficulties in the review.
We appreciate your time spent reviewing our manuscript as well as your criticisms and critiques of the reviewed manuscript. I agree with your opinion, line numbering was included in the manuscript.
- Please correct the usage of the degree Celsius throughout the abstract and other parts of the manuscript.
Ok, was done with all the manuscript.
- Please replace those from the title.
Ok, I agree with your opinion, done in the manuscript.
- BHT and TBHQ are safe to use in foods when added in the proper amount. The reported risks to health are mainly related to higher doses than that indicated by surveillance agencies. Please reword your sentence in order not to mislead the readers.
Ok, I agree with your opinion, done in the manuscript.
- This sentence has no sense: “Thence, the application of natural antioxidants extracted from plants and/or alga has received a lot of attention recently [6], essential oil [7], and sesame peel [8].”
The sentence is now clear.
- GRAS and other acronyms must be introduced in their first mention.
OK, written completely as generally recognized as safe (GRAS)
- Equations must be appropriately organized; some of them are totally disorganized and not adequately edited.
The equations are organized and included in the manuscript.
- Please standardize the use of SI units according to the most reported, e.g., “h” instead of “hr”; sometimes, the authors use (p≥0.05), sometimes (P≥0.05).
SI units and P≥0.05 were done in all manuscript.
- Section 2.6.4– Why do the authors performed Rancimat analysis at 100 °C when the AOCS indicates 110°C? Besides, the reference cited (which is outdated) indicates using three temperatures (100-120 °C), so anyone can predict the oil’s shelf life.
Ok, I agree with your opinion, already Rancimat analysis was done at 110 °C, this is typing mistake.
- Furthermore, shelf-life prediction based on Ref [39] using the Q10 coefficient must be adequately explained; it is not possible to understand how the authors calculated it using the induction period from a single temperature measurement.
The line slopes were measured by the plotting concentration and time was used to calculate the temperature acceleration factor (Q10), which was based on the increase in oxidation level for every 10 C increase in temperature using the following equation;
Q 10= e (T1-T2)/10
Where e is constant equal (2), T1 is the accelerated temperature (65 °C), and T2 is the ambient temperature (25 °C)
Q10= [2] (65-25)/10
Q10= [2] 40/10
Q10= [2] 4
Q10= 16 days
Thus, the acceleration factor was approximately 16 days, which means that one day at accelerated conditions (65 °C) is equivalent to 16 days at ambient (25 °C).
Prediction for the control sample which is stable for up to 12 days at 65 °C
Prediction control sample= 12x16= 192 days
Prediction for Ga-NPsE (600 ppm) which stable for up to 20 days at 65 °C
Prediction Ga-NPsE (600 ppm)= 20x16= 320 days
- The text must be proofread for grammar, as there are some mistakes and issues. For instance, “The organoleptic test of sunflower oil (SFO) was done”
Ok, done “The organoleptic test of sunflower oil (SFO) was done”
- In Section 3.1.1, if the authors mention all the mean values from the Tables, they must choose to present only some ranges in the text or eliminate the table. Presenting both is not reasonable.
Ok, I agree with your opinion, done in the manuscript of the ranges.
- Other sentences with no grammar sense: “The data revealed that garlic extract has a high antioxidant ability was 11.8 and 10.8 μg mL-1 for DPPH and ABTS, respectively.” “Results illustrated in Fig 1 confirm that garlic extracts a significant amount of organosulfur compounds.” Please proofread the manuscript.
Ok, I agree with your opinion, the sentences were corrected in the manuscript.
- The authors say that “From the abovementioned results, Ga-NPs could be employed as an antioxidant substitute in oils, fats, and fatty foods,” but I disagree. There is no how to reach such a conclusion using only two antioxidant potential methods mainly based on radical scavenging. Other chemical and biological-based antioxidant mechanisms, such as electron transfer or inhibition of lipid peroxidation, must be accomplished to say that properly. The tests performed support antiradical activity only.
Ok, I agree with your opinion, the sentence was deleted.
- Footnotes from Tables must be corrected to indicate clearly that lowercase/uppercase letters indicate significant differences.
All Footnotes are clear now in all tables
abc: There is no significant differences between any two means 'in the same column' have the same lowercase superscript letter (P≥0.05).
ABC: There is no significant differences between any two means ‘in the same row’ have the same uppercase superscript letter (P≥0.05).
- Tables 2, 3, and 4 could be merged into one, as their data all represent quality indicators. There is no mention of what the R2 represents. Also, there is no reasonability of an R2 equal to 1 (Ga-NPsE (1000 ppm)). In such cases, the authors may increase decimal digits. The authors did not standardize the number of significant digits in some tables. Include the concentration of BHT and a-tocopherol in the tables.
We find it difficult to merge the tables together into one, R2 was deleted. All corrections were done in the manuscript.
- Clearly, Figure 1 has been produced based on the print screen, as some non-recognized words have been marked. The authors must edit properly and provide better-quality images. The resolution is too low.
Ok, done
- In section 3.2.1, the authors commented on peroxide values, but there is no mention of the standards of such parameters for oils. Are the samples within the standard limits recommended by the regulatory agencies? Also, what do the authors mean with “GaNPsE 1000 ppm had R2=1, after 24 days of storage”?
The standard of peroxide value was placed in the manuscript, and R2 was deleted
- The authors state that the p-anisidine value “slightly increased from 4.20 to 25.26 mg kg-1 until day 24 of storage”, but such an increase is not slight; it is significantly high. The following conclusion stated by the authors is also controversial, as the data indicates that the antioxidants were able to reduce but not avoid the increase in the p-AV. Be careful with such conclusions so as not to mislead the readers; the Tables disagree with the discussion in this regard.
Ok, I agree with your opinion, all corrections were done in the manuscript.
- Another sentence with no sense: “Moreover, R2 = 1 for Ga-NPsE (1000 ppm) and 0.89 for control.”
This sentence was deleted
- In the following sentence, “The findings agree with those reported by Carelli et al. [53],” what those authors concluded that your findings agree with? Please be specific, as the manuscript is full of such sentences without proper contextualization.
Ok, clear now in the manuscript
- Another sentence with no sense: “No significant differences in antioxidant ability (AA) scores between samples containing GaNPsE 1000 ppm and BHT (P≥0.05).”
Ok, clear now in the manuscript
- Charts from Figure 2 must be appropriately placed side-by-side for better visualization. In fact, they seem to represent the same data from the Tables. If so, there is no need for such figures. Please double-check.
Ok, the charts are placed side-by-side.
- In the same way, charts from Figure 3 must be appropriately placed side-by-side for better visualization and to avoid splitting them into different pages.
Ok, the charts are placed side-by-side.
- There is no suitable discussion and contextualization of the sensory scores,
Ok, the discussion and contextualization of the sensory scores are now clear and suitable
Fig. 3 depicts the sensory parameters, such as color, aroma, and acceptability, of sunflower oil (SFO) treatments during storage. The sensory assessment of SFO samples was done for different storage periods due to the ending of shelf life with the intelligible development of off-flavor. Panelists generally approved of all sensory properties of SFO treatments at time zero, as the storage progressed, the sensory acceptability decreased. The control sample scored the lowest color value during storage periods, while the oil sample incorporated Ga-NPS at different levels was accepted up to day 24. The aroma value differed significantly (P≤0.05) between the oil treatments and the control group. In SFO samples incorporated BHT or α-tocopherol or Ga-NPS 1000 ppm were accepted until day 24 of storage, however, the control was rejected on day 12 due to off-flavor. SFO samples' acceptability values decreased significantly over time, whereas BHT and Ga-NPS 1000 ppm samples did not differ after 24 days. The sensory data demonstrated that the control sample was rejected after day 12 of storage; however, the oil samples incorporated BHT, α-tocopherol, and Ga-NPS 1000 ppm were accepted up to day 24. These changes match well with chemical indicators i.e. PV, p-AnV, and TV. These findings are consistent with those noted by Iqbal and Bhanger [45] who found that the chemical and sensory parameters an important to evaluate the efficacy of garlic antioxidants in sunflower oil.
- This sentence brings all the information from Figure 4 and should be revised, or Figure 4 removed: “Nevertheless, the expected shelf life of SFO at 25 °C was ∼192, 256, 320, 384, 256, 384, and 384 days, respectively.”
Ok, the sentence was revised
- Conclusion section. The authors cannot infer such a conclusion that Ga-NPsE can be used instead of BHT with such preliminary data based on 24 days of storage at a single temperature; many more tests must be performed to reach such an exact conclusion. They could merely suggest that such Ga-NPsE may be an alternative antioxidant but indicate that more tests must be assessed to prove such behavior better.
Ok, I agree with your opinion
- The following sentence is not a conclusion, but material and methods: “Ga-NPsE were placed into SFO at three levels i.e. 200, 600, and 1000 ppm (w/v), and compared with 600 ppm garlic lyophilized powder extract (Ga-LPE), 200 ppm BHT, and 200 ppm α-tocopherol and without Ga-NPsE (control).”
Ok, I agree with your opinion
General comment:
- Nowadays, the authors must be aware that the first step in the publication process is to provide a high-quality document to the reviewers, so we can focus on suggesting improvements to the main content of the study, other than the language, writing style, figures, tables, and format.
Ok, thank you
In the next manuscript, we will be keen to provide a high-quality manuscript and take into account all your comments and guidance. We appreciate the time spent reviewing our manuscript.
Again, thank you for the opportunity to submit our manuscript to Foods Journal. If I can be of additional assistance with regard to this submission, please get in touch with me directly at +2 0122 3831825 or via email at mohamed.abdelhafez@fagr.bu.edu.eg
Regards,
Mohamed K Morsy, Ph. D.
Associate Professor at Food Technology Dept.

Round 2
Reviewer 2 Report
- I asked the authors to replace the keywords that are mentioned in the title to different ones, not to use the same words from the title as keywords.
- Chromatogram images are still with the same low quality and typo problems indicated previously
- Title of the Y-axis in Figure 2 must be corrected
- Footnotes in the tables regarding the significant differences have grammar issues and lack coherence
- The authors must verify if there are any correlations (Person's) between the TPC, TFC and antioxidant activity (Table 1) and include a brief discussion in the text.
Author Response
To Whom It May Concern;
Enclosed is the revised manuscript entitled “QTRAP LC/MS/MS of Garlic Nanoparticles and Enhancing Sunflower Oil's Oxidative Stability under Accelerated Storage”.
We thank the Scientific Editor and all the reviewers for their comments and suggestions, as they have significantly improved the manuscript. The following are responses to the reviewers' comments. We hope that the manuscript is acceptable after these revisions.
Reviewers' comments
Reviewer # 2
- I asked the authors to replace the keywords that are mentioned in the title to different ones, not to use the same words from the title as keywords.
We appreciate your time spent reviewing our manuscript as well as your criticisms and critiques of the reviewed manuscript. We replaced the keywords in the title as follows “QTRAP LC/MS/MS of Garlic Nanoparticles and Improving Sunflower Oil Stabilization during Accelerated Shelf-life Storage”
- Chromatogram images are still with the same low quality and typo problems indicated previously.
The chromatogram images became a clear and high quality
- Title of the Y-axis in Figure 2 must be corrected
Ok, done in the manuscript.
- Footnotes in the tables regarding the significant differences have grammar issues and lack coherence.
Ok, done as follows
There are no significant differences between any two means in the same column that have the same lowercase superscript letter (P≥0.05).
There are no significant differences between any two means in the same row that have the same uppercase superscript letter (P≥0.05).
- The authors must verify if there are any correlations (Person's) between the TPC, TFC and antioxidant activity (Table 1) and include a brief discussion in the text.
Ok, done in the manuscript.
The analysis of the data revealed that the antioxidant activity of garlic cultivars was correlated with TPC and TFC (R=1). The TFC and TPC became increasingly important to antioxidant activity. [44, 45] also found that antioxidant ability was positively correlated with TFC and TPC.
Again, thank you for the opportunity to submit our manuscript to Foods Journal. If I can be of additional assistance with regard to this submission, please contact me directly at +2 0122 3831825 or via email at mohamed.abdelhafez@fagr.bu.edu.eg
Regards,
Mohamed K Morsy, Ph. D.
Associate Professor at Food Technology Dept.